# Effectiveness of Non-Local Means Algorithm with an Industrial 3 MeV LINAC High-Energy X-ray System for Non-Destructive Testing

**DOI:** 10.3390/s20092634

**Published:** 2020-05-05

**Authors:** Kyuseok Kim, Jaegu Choi, Youngjin Lee

**Affiliations:** 1Electro-Medical Device Research Center, Korea Electrotechnology Research Institute (KERI), 111, Hanggaul-ro, Sangnok-gu, Ansan-si, Gyeonggi-do 15588, Korea; kskim502@keri.re.kr (K.K.); jgchoi@keri.re.kr (J.C.); 2Department of Radiological Science, Gachon University, 191, Hambakmoero, Yeonsu-gu, Incheon 21936, Korea

**Keywords:** non-destructive testing, industrial high-energy X-ray imaging system, denoising algorithm, non-local means approach, quantitative evaluation of image performance

## Abstract

Industrial high-energy X-ray imaging systems are widely used for non-destructive testing (NDT) to detect defects in the internal structure of objects. Research on X-ray image noise reduction techniques using image processing has been widely conducted with the aim of improving the detection of defects in objects. In this paper, we propose a non-local means (NLM) denoising algorithm to improve the quality of images obtained using an industrial 3 MeV high-energy X-ray imaging system. We acquired X-ray images using various castings and assessed the performance visually and by obtaining the intensity profile, contrast-to-noise ratio, coefficient of variation, and normalized noise power spectrum. Overall, the quality of images processed by the proposed NLM algorithm is superior to those processed by existing algorithms for the acquired casting images. In conclusion, the NLM denoising algorithm offers an efficient and competitive approach to overcome the noise problem in high-energy X-ray imaging systems, and we expect the accompanying image processing software to facilitate and improve image restoration.

## 1. Introduction

In industrial inspection, X-ray imaging is well established for non-destructive testing (NDT), and it is used for crack detection, assembly inspection, analysis of material composition, etc. [1,2,3]. Depending on the specimen, industrial radiography systems are often affected by deterioration of image quality when radiation penetrates high-density objects. Attempts to overcome the penetration of dense objects led to the development of the industrial high-energy X-ray imaging system, which is now being widely used. In particular, low- and high-energy radiation are generally distinguished by considering energy below 1 MeV as low energy, whereas a high-energy imaging system would operate in the range of 1–9 MeV [4,5,6].

X-ray image degradation is a crucial issue, and it is mainly contaminated by noise through the imaging system. Thus, the recovery of a deterioration of image performance, collectively referred to as image denoising, has been extensively researched. The correlated noise of X-ray statistical fluctuation is considered as the Poisson distribution and the uncorrelated noise with X-rays. In particular, the uncorrelated noise is mainly generated from the electric fluctuation, and it is considered as Gaussian distribution [7]. In context with the characteristics of the X-ray image, most of the image denoising methods depend on the X-ray image degradation model [8,9], which is assumed to be linear and shift-invariant and may be described as follows:(1)$$g\left(x\right)=f\left(x\right)+\eta \left(f\right)\delta ,$$
where $g\left(x\right)$ is the X-ray image acquired from the imaging system in pixel position *x*, $f\left(x\right)$ is the original (i.e., noise-free) image, $\eta \left(f\right)$ is the standard deviation of the noise distribution, and δ is the Gaussian noise (with zero mean and a standard deviation equal to one). In this noise model, noise variance, $\eta^{2}\left(f\right)$, can be separated into Poisson and Gaussian components as:(2)$$\eta^{2}\left(f\right)=Pf\left(x\right)+G^{2},$$
where P and G are the Poisson noise parameter and the standard deviation of the Gaussian noise, respectively.

The deterioration of image quality due to noise in industrial high-energy X-ray images is a crucial issue in NDT. Reducing the noise that affects NDT images is indispensable to improve the accuracy of inspection. Common filtering methods involve the use of an average filter or a median filter [10]. The former of these two filters is mainly implemented by compressing Gaussian and uniform noise, whereas the latter filter compensates for salt-and-pepper noise [11]. A more advanced method was proposed by Wang et al., who developed a noise reduction algorithm incorporating a wavelet transform, which entails multi-resolution analysis [12]. This method is based on a new type of directional adaptive median filter designed to reduce the amplitude of the noise. However, X-ray images are also adversely affected by Poisson noise, which is associated with X-ray photons, and Gaussian noise, which is related to electronic noise, and both of these types of noise are widely distributed throughout X-ray images [7]. Therefore, a suitable method is required to deal with each of these types of noise. An alternative solution to this problem is the Anscombe transform (AT) [13,14], which is effective for converting Poisson noise to Gaussian noise. However, these techniques are computationally expensive because of the calculation of a complex optimization problem, and they often behave according to the use of non-convex models. Moreover, it is difficult to preserve the edge smoothing in the acquired X-ray images. To overcome this problem, various noise reduction algorithms have been proposed and are widely used as state-of-the-art methods, including gradient models [15] and non-local self-similarity models [16,17]. These methods, which are typically based on iterative algorithms, include total variation (TV) and non-local means (NLM). In particular, the advantages of the NLM method are its superior performance in terms of image quality and processing time as well as its ability to maintain the edge structure of images. Notably, Xu et al. [17] introduced the NLM approach by modeling the non-local self-similarity (NSS) prior to noise reduction in a natural image. However, its approach has limited applications in 2D X-ray imaging, including for Poisson–Gaussian mixed noise due to the assumption that it only considers the Gaussian noise. Since Gaussian noise is independent of the imaging system, it is possible to artificially generate and learn various forms of Gaussian noise in advance. On the other hand, Poisson noise depends on the photon fluence (amount of photons). In addition, the X-ray image is expressed by superimposing the image from 3D to 2D, which has different characteristics from the natural image. It is difficult to mathematically understand that there are many similar patches in an X-ray image, notwithstanding overlap and ambiguous borders compared to that of the natural image.

The objective of this paper is to propose an NLM denoising algorithm and evaluate the image quality obtained with this algorithm using the industrial high-energy X-ray imaging system we established. Our proposed method does not require additional manual work and a large amount of prior information for the removal of the mixed noise component, effectively. In this study, we designed a Wiener filter and TV denoising algorithm to enable us to compare our proposed NLM algorithm with existing algorithms. We evaluated the noise reduction performance by visual assessment and using the intensity profile, root-mean-square error (RMSE) [18], edge preservation index (EPI) [19], contrast-to-noise ratio (CNR) [20], coefficient of variation (COV) [21], and normalized noise power spectrum (NNPS) [22,23] of three phantom images.

## 2. Materials and Methods

### 2.1. Modeling the Proposed NLM Denoising Algorithm 

The Wiener filter, which can effectively reduce noise, is well known as a mean-square error-based optimal stationary linear filter for image restoration. The transfer function of the Wiener filter [24] in the Fourier domain is given by:(3)$$W\left(u,v\right)=\frac{S_{f}\left(u,v\right)}{S_{f}\left(u,v\right)+\sigma^{2}},$$
where $S_{f}\left(u,v\right)$ is the power spectrum of the original image and $\sigma^{2}$ is the variation of the noise image with zero mean. The known limitation of this method is that it is prone to over-smoothing [25]; therefore, another approach is to introduce the noise parameter $\sigma_{n}$ as follows:(4)$$\tilde{f}\left(x,y\right)=m_{f}\left(x,y\right)+\frac{\sigma_{f}^{2}\left(x,y\right)}{\sigma_{f}^{2}\left(x,y\right)+\sigma_{n}^{2}\left(x,y\right)}\left(g\left(x,y\right)-m_{f}\left(x,y\right)\right),$$
where $m_{f}\left(x,y\right)$ is the local mean value of the original image, $\sigma_{f}^{2}\left(x,y\right)$ and $\sigma_{n}^{2}\left(x,y\right)$ are the local variations of images, $g\left(x,y\right)$ is the obtained image with the noise signal, and $\tilde{f}\left(x,y\right)$ is the restored image. TV regularization applies image denoising while preserving important details, a concept that was introduced by Rudin, Osher, and Fatemi (i.e., the ROF model):(5)$$\emptyset \left(f\right)=\underset{f\in \Omega }{\mathrm{argmin}}\|f{\|}_{TV}+\frac{\lambda }{2}\sum_{\Omega }{\left(g-f\right)}^{2},$$
(6)$$\emptyset \left(f\right)=\underset{f\in \Omega }{\mathrm{argmin}}\sum_{\Omega }\left[\|\nabla f\|+\frac{\lambda }{2}{\left(g-f\right)}^{2}\right],$$
(7)$$\|\nabla f\|=\underset{x,y}{\sum }\left[\left|f_{x+1}-f_{x}\right|+\left|f_{y+1}-f_{y}\right|\right],$$
where f is the exact image without noise in Cartesian coordinates *x* and *y*, g is the degraded image, Ω is the boundary condition, and λ is a balancing parameter between the fidelity and regularization terms. Although this method performs effectively with respect to noise reduction and simultaneously preserves edges, it has a tendency to over-flatten the degraded image and produce a so-called cartoon-like artifact when the result is deduced using a high balancing parameter in the regularization term [26]. The NLM method, which delivers remarkable performance with regard to reducing the noise, as shown in Figure 1, is represented by the following simple formula:(8)$$NL\left(g\left(i\right)\right)=\frac{1}{Z\left(i\right)}\sum_{j}w\left(i,j\right)g\left(i\right),$$
(9)$$w\left(i,j\right)=exp\left(-\frac{p\left(i,j\right)}{d^{2}}\right),$$
(10)$$p\left(i,j\right)=\frac{1}{\delta }\|g\left(\delta_{i}-\delta_{j}\right){\|}_{2}^{2},$$
where i is a pixel, which is calculated by averaging the weighted pixels, and $w\left(i,j\right)$ is a family of weights that depend on the similarity between pixels i and j, and satisfy the usual conditions 0≤
$w\left(i,j\right)\leq 1$ and $\underset{j}{\sum }w\left(i,j\right)=1$. Further, $Z\left(i\right)$ is the normalized constant of the similarity between two square patches between $\delta_{i}$ and $\delta_{j}$ at center pixel i and j, and d is a filtering parameter, which is the standard deviation of the exponential function, as illustrated in Figure 1. The algorithm of the NLM method operates with respect to local filters; in other words, it not only compares the pixel intensities, but also the geometrical configurations in an entire neighborhood [27]. This, along with the fact that it does not introduce additional artifacts while minimizing the loss of the edge structure in the image, is the reason for the successful noise reduction capabilities of this method.

### 2.2. Industrial High-Energy X-ray Imaging System and Materials Modeling

We constructed a high-energy X-ray system for industrial inspection. The system is composed of a linear accelerator (LINAC, HEXTRON3/1-500, Granpect Inc.), a large-area flat-panel detector (pixel size: 200 μm, pixel matrix: 2048 (W) × 2048 (H), Gd_2_O_2_S scintillator, 16 bit A/D), and a mechanical support for object installation, as shown in Figure 2. The distance of source to object is 800 mm, and the distance of object to detector is 80 mm. The exposure time was about 5 s to obtain a single image. The additional specifications of the 1−3 MeV LINAC X-ray generator are listed in Table 1.

Figure 3a–c shows an aluminum die-casting, a concrete dummy, and a 50 mm Pb plate phantom with image quality indicator (IQI) penetrameters (ASTM E1742) that were used in the experiment, respectively. Here, the hole-type penetrameters have a thickness of approximately 2% of that of the specimen based on the American Society for Testing and Materials (ASTM) testing protocol.

### 2.3. Evaluation of Image Quality

We evaluated the performance visually and used the intensity profile to assess the improvement in the image quality of X-ray phantom images based on our designed algorithms. In addition, the RMSE, EPI, CNR, COV, and NNPS parameters were used to quantitatively compare and evaluate the performance of the conventional algorithm and the proposed NLM algorithm for X-ray phantom images. These parameters were calculated as follows:(11)$$\mathrm{RMSE}=\sqrt{\frac{\sum_{i=1}^{N}{\left(I_{1}-I_{2}\right)}^{2}}{N}},$$
(12)$$\begin{array}{c} \mathrm{EPI}=\frac{\Gamma \left(⊿I_{1}-\bar{⊿I_{1}},⊿I_{2}-\bar{⊿I_{2}}\right)}{\sqrt{\left(⊿I_{1}-\bar{⊿I_{1}},⊿I_{1}-\bar{⊿I_{1}}\right)}\circ \Gamma \left(⊿I_{2}-\bar{⊿I_{2}},⊿I_{2}-\bar{⊿I_{2}}\right)},\\ \left(\Gamma \left(a,b\right)=\sum_{x_{i},y_{j}\in ROI}I_{1}\left(x_{i},y_{j}\right)I_{2}{\left(x_{i},y_{j}\right)}^{2}\right) \end{array}$$
(13)$$\mathrm{CNR}=\frac{\left|S_{\mathrm{Target}}-S_{\mathrm{Background}}\right|}{\sqrt{\sigma_{\mathrm{Target}}{}^{2}+\sigma_{\mathrm{Background}}{}^{2}}},$$
(14)$$\mathrm{COV}=\frac{\sigma_{\mathrm{Target}}}{S_{\mathrm{Target}}},$$
(15)$$\begin{array}{c} \mathrm{NNPS}\left(u,v\right)=\frac{\mathrm{NPS}\left(u,v\right)}{{\left(\mathrm{large}\;\mathrm{area}\;\mathrm{signal}\right)}^{2}},\\ \;\left(\mathrm{NPS}\left(u_{n},v_{k}\right)=\underset{N_{x}N_{y,}M\to \infty }{\lim}\frac{\Delta x\Delta y}{M\cdot N_{x}N_{y}}\sum_{m=1}^{M}|\sum_{i=1}^{N_{x}}\sum_{j=1}^{N_{y}}\left(I\left(x_{i},y_{j}\right)-S\left(x_{i},y_{j}\right)\right)\exp\left(-2\pi i\left(u_{n}x_{i}+v_{k}y_{i}\right)\right){|}^{2}\right), \end{array}$$
where $I_{1}$ is reference data, $I_{2}$ is measured data, N is number of matrix elements, and $\bar{⊿I}$ is operation of Laplacian filtering in the region of interest (ROI). $S_{\mathrm{Target}}$ and $\sigma_{\mathrm{Target}}$ are the mean and standard deviation of the intensity in the target ROI, respectively; $S_{\mathrm{Background}}$ and $\sigma_{\mathrm{Background}}$ are the mean and standard deviation for the background intensity in the ROI, respectively; S is the average background intensity; and $N_{x}N_{y}$ and Δx,Δy account for the pixel numbers and pixel sizes on the *x*- and *y*-axes, respectively.

## 3. Results and Discussion

Figure 4 shows the simulation results of the original image (noise-free), artificial noisy component, noisy image, restored image with Wiener filtering, restored image using the total variation, and restored image using the NLM algorithm using numerical phantom. We implemented the X-ray simulation based on the ray-tracing method. Here, the computational language MATLAB^TM^ (Mathworks, USA, R2018b) was used and the original image was acquired in the same geometry condition as the experiment. Moreover, we generated the noise component, which was composed of Poisson and Gaussian distributions, using the *imnoise* (·) and *poissrnd* (·) functions (i.e., we used the parameters: Poisson parameter = 2.0 and Gaussian parameter = 2.0) in a MATLAB toolbox based on Equations (1) and (2). The image restored using the NLM algorithm was slightly closer to the original image, compared to that of Wiener filtering and TV-based denoised image. For the quantitative evaluation, we measured the RMSE and EPI values (Table 2). The RMSE values of the noisy image and denoised images of the Wiener filter, TV, and NLM were approximately 38.7, 12.5, 6.5, and 1.2, respectively. A smaller RMSE value corresponds to a greater similarity to the pixel value of the original image without additional degradation. The EPI factors of the images were approximately 0.17, 0.3, 0.74, and 0.87, respectively. An EPI value closer to 1 shows that the edge of the measured image is well-preserved.

Figure 5 shows the images of an aluminum die-casting phantom that were acquired with our established high-energy X-ray imaging system and processed using various denoising methods, including the proposed NLM algorithm and were ultimately used to obtain the intensity profile. The qualitative analysis we conducted confirmed the ability of the proposed algorithm to recover a blurred artifact from only the noise-reduced image. A visual assessment of the phantom image, including the magnified regions, indicates that the noise distribution in the phantom image is clearly reduced using the proposed NLM denoising algorithm. In particular, it was confirmed that the edges in the NLM-based denoised image are emphasized compared to those in the TV-based image, and distortion due to contrast amplification is reduced with our proposed algorithm. In addition, Figure 6 shows the measured intensity profile of the noisy image and the noise-reduced images and shows that the signal intensity fluctuation of the profile for the proposed NLM-based denoised image is smaller than that of the other denoising algorithms.

Figure 7 shows images of the concrete dummy phantom that were acquired with the high-energy X-ray imaging system we developed. The images were processed using various denoising methods, including the proposed NLM algorithm with an ROI to evaluate the CNR and COV. The evaluated CNR and COV results for the denoising algorithms are shown in Figure 8. The CNR obtained with the NLM algorithm was approximately 17.2, 4.5, and 3.4 times higher than those obtained with the noisy image, Wiener filter, and TV denoising algorithm, respectively. In contrast to the CNR results, the COV increased in the following order, and the COV obtained with the NLM algorithm was approximately 28.4, 10.9, and 3.4 times smaller than those obtained with the noisy image, Wiener filter, and TV denoising algorithm, respectively. Based on the equations, the higher the CNR result and the smaller the COV value, the higher the quality of the image. Overall, these results confirmed that, compared to the contrast, the value of the signals of the target region in the phantom image was affected to a greater extent by the denoising algorithm or filter. However, the difference between the CNR and COV of the TV denoising algorithm and our proposed NLM algorithm was 3.4 times, which means that the algorithm had a similar effect on the contrast and the signal. Figure 9 shows the images of a 50 mm Pb plate phantom that were acquired with the high-energy X-ray imaging system we established. The images were processed using various denoising methods, including the proposed NLM algorithm with an ROI_A_ to evaluate the NNPS. The NNPS results calculated using the various noise reduction techniques are shown in Figure 10. In all cases, it was confirmed that the overall NNPS value decreased as the spatial frequency increased. The NNPS for the proposed NLM algorithm is the lowest (approximately 10^−10^ mm^2^ at 2.0 lp/mm spatial frequency) and approximately 10^5^ times lower than that of the noisy image. These results show that the proposed NLM algorithm provides a significantly larger reduction of the noise intensity, which improves the noise performance in all spatial frequency areas. In addition, the noise distributions of the Wiener filter and TV denoising algorithm in the X-ray phantom images showed similar tendencies as the spatial frequency increased. In particular, the TV denoising algorithm showed that the NNPS value for a spatial frequency of approximately 1.0 lp/mm either remained stable or increased. These results indicate that the algorithm was not effectively applied in the high-frequency domain. As a result, the proposed NLM algorithm was highly efficient in the high-frequency domain, is sensitive to noise, and proved its usefulness when processing high-energy X-ray images.

In industrial applications, noise reduction for high-energy X-ray imaging is highly important to improve the image quality. The high-energy X-ray imaging system we established achieved superior noise reduction performance with the NLM denoising algorithm in terms of all evaluation parameters. Our results demonstrate that this algorithm provides a bright future for industrial high-energy X-ray imaging systems with excellent image quality, which has considerable potential in NDT applications. Based on the results of this study, we expect it to become possible to apply the technique to industrial X-ray images of higher energy in the future.

## 4. Conclusions

We established an industrial high-energy X-ray imaging system using a linear accelerator and large-area flat-panel detector and designed an NLM denoising algorithm, which was employed to improve the quality of various phantom images acquired with the system. We implemented the proposed process to improve the image quality and performed systematic experiment to evaluate the imaging characteristics in terms of the intensity profile, CNR, COV, and NNPS. The image characteristics of the restored image were significantly improved; the CNR value of the NLM algorithm was approximately 17.2 and 3.4 times higher than that of the noisy and TV images, respectively. Meanwhile, the COV value of the NLM algorithm was approximately 28.4 and 3.4 times smaller than that of the noisy and TV images, respectively. Moreover, the NNPS value at a spatial frequency of 2 lp/mm was about 1 × 10^−10^ mm^2^, about 10^5^ times lower than that for the noisy image. Our results demonstrated that the proposed algorithm is highly suitable for application in an industrial high-energy X-ray imaging system and can be used to significantly enhance the quality of the acquired images.

## Figures and Tables

**Figure 1 sensors-20-02634-f001:**
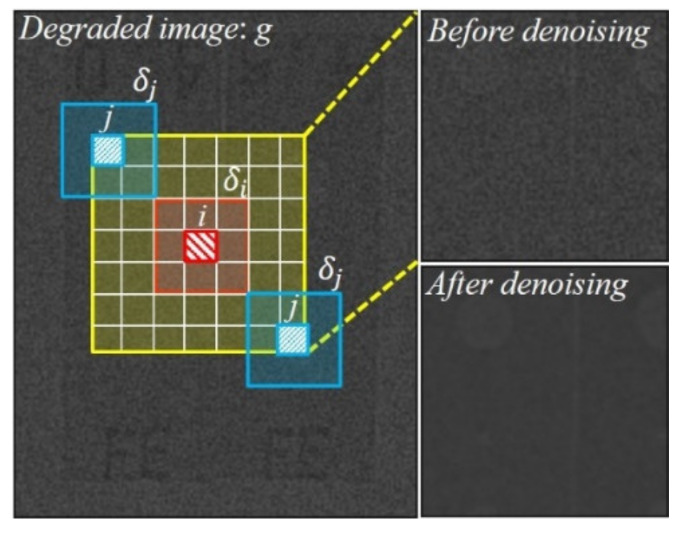
Schematic illustration of non-local means method for effective noise reduction in high-energy X-ray images.

**Figure 2 sensors-20-02634-f002:**
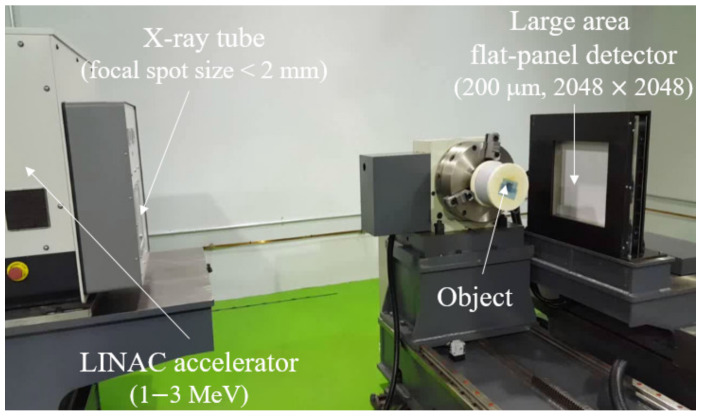
Photographic image of a 1−3 MeV energy X-ray system consisting of an X-ray tube (focal spot size 2 mm) with a linear accelerator (LINAC) and large-area flat-panel detector (pixel size = 200 μm, 2048 × 2048).

**Figure 3 sensors-20-02634-f003:**
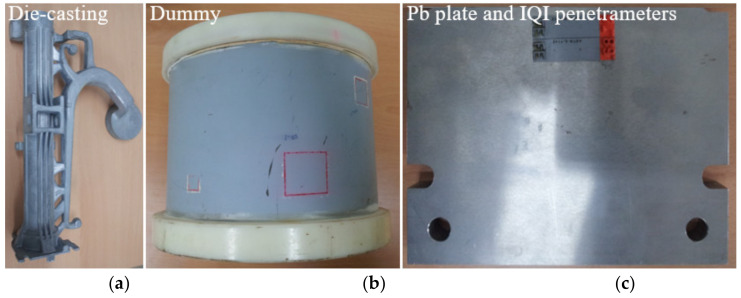
Photographic images of the (**a**) aluminum die-casting, (**b**) concrete dummy, and (**c**) 50 mm Pb plate attached to the image quality indicator (IQI) penetrameters for the experiment.

**Figure 4 sensors-20-02634-f004:**
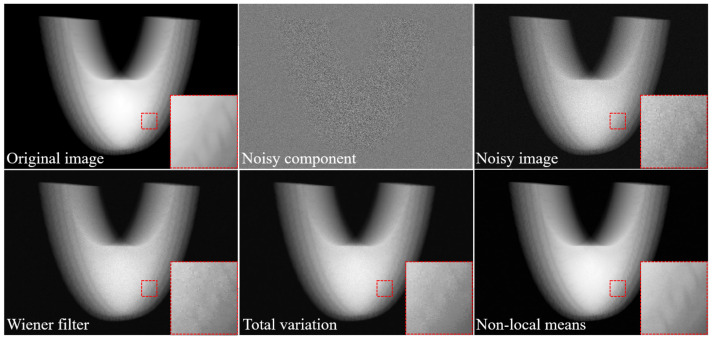
Examples of the simulation study: original, noise component, Wiener filter, total variation, and proposed non-local means denoising algorithm, acquired with our simulation modeling and these magnified images in a selected region.

**Figure 5 sensors-20-02634-f005:**
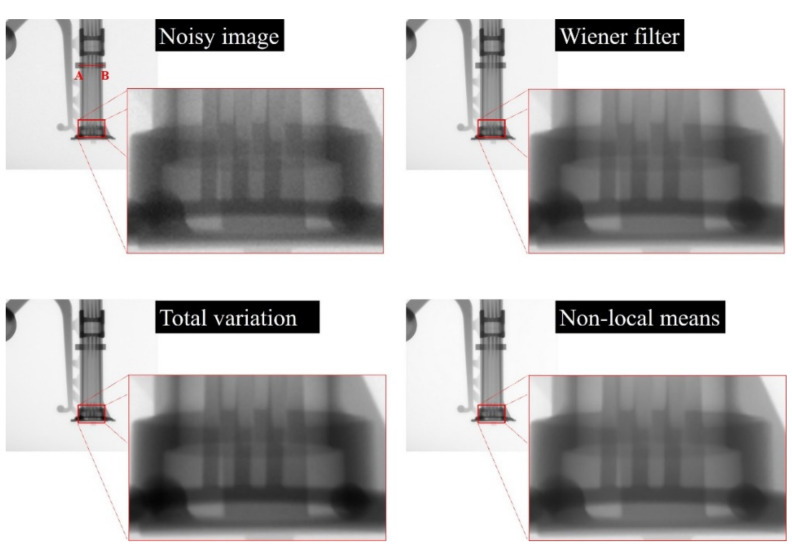
Images of the aluminum die-casting phantom: noisy, Wiener filter, total variation, and proposed non-local means denoising algorithm, acquired with our established industrial high-energy X-ray imaging system. A magnification of a selected region is shown for visual assessment.

**Figure 6 sensors-20-02634-f006:**
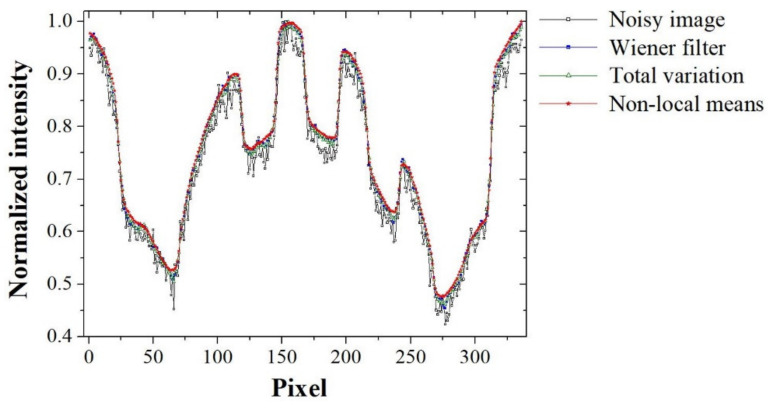
Intensity profile (using line AB in Figure 5) in relation with denoising methods applied to the acquired image of the aluminum die-casting phantom.

**Figure 7 sensors-20-02634-f007:**
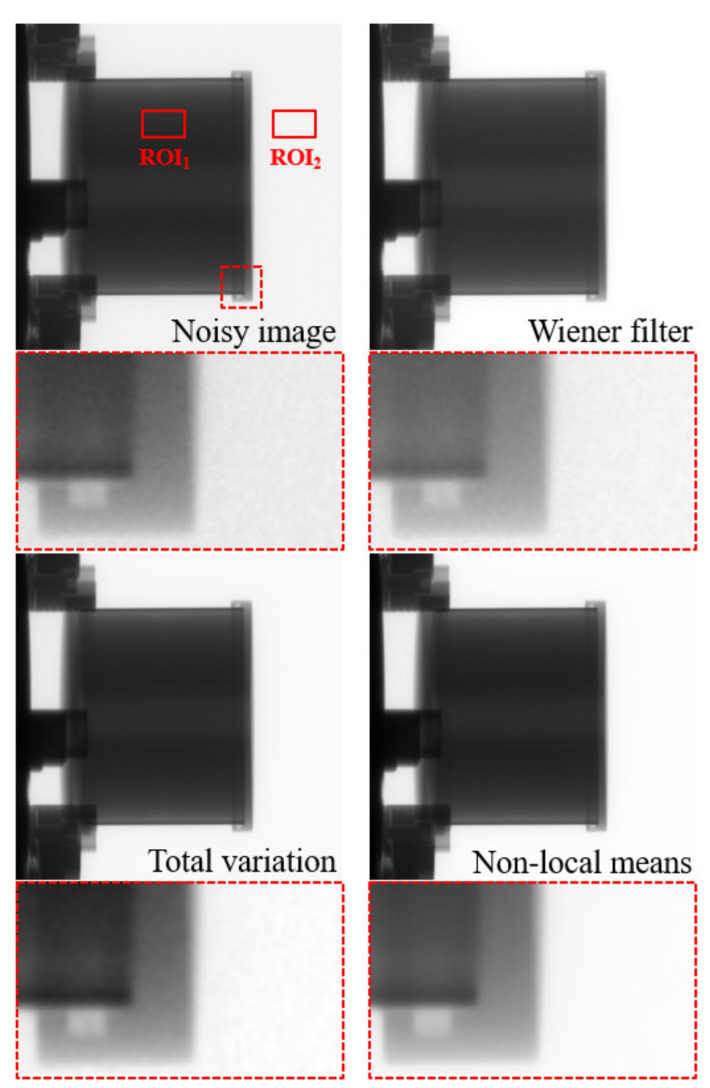
Images of the concrete dummy phantom: noisy, Wiener filter, total variation, and proposed non-local means denoising algorithm, acquired with our established industrial high-energy X-ray imaging system. The contrast-to-noise ratio and coefficient of variation were calculated for the specified regions of interest (ROI_1_ and ROI_2_).

**Figure 8 sensors-20-02634-f008:**
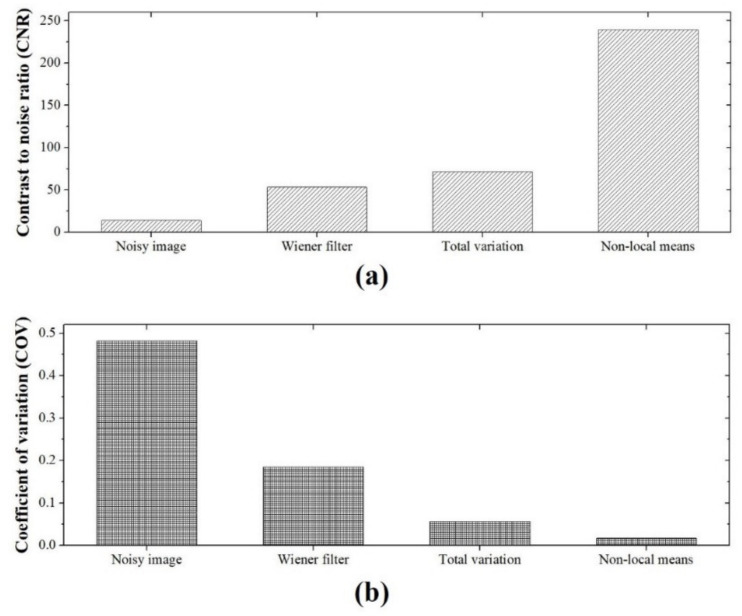
Experimental results for the (**a**) contrast-to-noise ratio (using ROI_1_ and ROI_2_ in Figure 7) and (**b**) coefficient of variation (using ROI_1_ in Figure 7) as a function of the denoising method using the image of the concrete dummy phantom.

**Figure 9 sensors-20-02634-f009:**
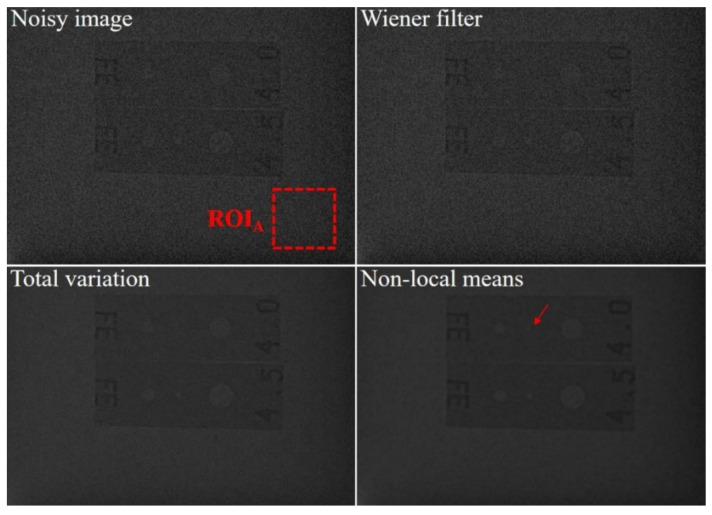
Images of the 50 mm Pb plate phantom with IQI penetrameters: noisy, Wiener filter, total variation, and proposed non-local means denoising algorithm. The images were acquired with our established industrial high-energy X-ray imaging system. The normalized noise power spectrum was calculated for the specified ROI_A_.

**Figure 10 sensors-20-02634-f010:**
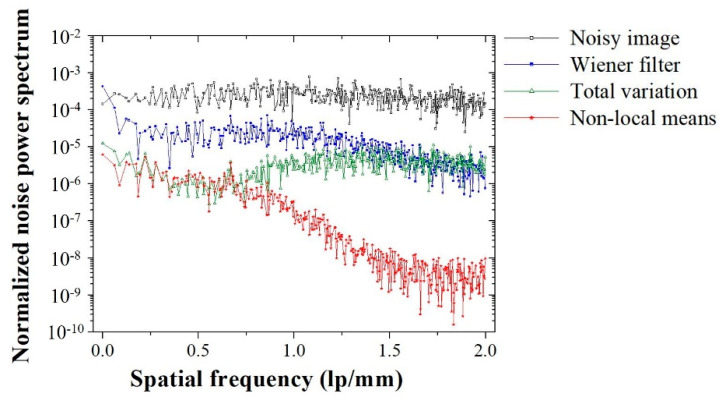
Normalized noise power spectrum (using the ROI_A_ in Figure 9) obtained by applying the different denoising methods to images that were acquired of the 50 mm Pb plate phantom with IQI penetrameters.

**Table 1 sensors-20-02634-t001:** Specifications of the X-ray generator at 3 MeV.

Categories	Values
Focal spot size	≤2 mm
Beam angle	0–23°
Beam symmetry	≤±5%
Dose rate	3.0 Gy/min@1m

**Table 2 sensors-20-02634-t002:** The results of root-mean-square error (RMSE) and edge preservation index (EPI) with respect to noise reduction method.

Noise Reduction Method	RMSE	EPI
Noisy image	38.7	0.17
Wiener filter	12.5	0.3
Total variation	6.5	0.74
Non-local means	1.2	0.87

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
