# Peer review of "Effectiveness of Non-Local Means Algorithm with an Industrial 3 MeV LINAC High-Energy X-ray System for Non-Destructive Testing"

_sensors, 2020, doi:10.3390/s20092634_

Round 1

Reviewer 1 Report

The authors proposed a denoising algorithm and presented three experimental images to demonstrate the superiority of their algorithm compared to Wiener filter and a TV algorithm. The result looks promising but I have some comments as follows:

  1. Is this paper about designing of the high-energy system or about the algorithm? It feels like the authors want to cover both sides but neither is not detailed enough. If they want to talk about their new system, then more quantitative discussions about the imaging quality and hardware needs to be presented, for example, resolution, exposure time, etc.

If it is about the algorithm, I expect more though comparison with previous NLM algorithms. Why is this proposed one better than the one, for example, in Ref 17, better? Why not use simulated images to compare with noise-free data to see the actual error?

  1. It is impossible to see the image quality difference in Figures 5, 7,and 9.
  2. Table 2 and 3 are unnecessary since these parameters are not discussed in the result section.
  3. There is no red arrow in fig 5
  4. Title is unnecessarily long.
  5. Some language needs to be modified. They are difficult to understand. For example: line 34 “suffered from radiation transmission of high density object”.
  6. Line 42-43.
  7. Line 74-75.

Reviewer 2 Report

This is a worthwhile paper that presents the application of a non-local means (NML) algorithm to high energy X-ray images. However, it seems rather confused as it falls across two areas that don’t seem to fit together well. To me the main topic of the paper is the application of the NLM algorithm, as this is where the majority of the technical discussion and experimentation lies. Other parts focus on hardware of the linac imaging system and seem out of place. This is because the first three pages, including the Abstract, Introduction and the majority of the Materials and Methods, hardly mentions the experimental imaging system and exclusively concentrates on image noise and the NLM algorithm, as does the last five pages.

Relevant characteristics of the imaging system are presented in section 2.2, along with a detailed account of the operation of the linac that seems out of place, especially as it appears to be a commercial instrument. There is no further discussion of the linac system and as already mentioned the remainder of the paper compares the NLM algorithm to other filters. So I suggest the linac sections (lines 143 to 152 and figure 3) are removed so the paper as a whole can concentrate on imaging and noise filtering.

The choice of test objects should be explained as they appear to have been chosen at random, which I’m sure isn’t the case. So a few lines should be included to explain why each one was used.  I’ve noted that the lead plate had two IQIs that are shown in figure 4 to be immediately under the top hole. While these don’t appear to have been used for any of the analysis the IQIs are still referred to through the rest of the paper, which is confusing. In figure 9 the NNPS is shown to have been calculated from a region on the opposite side of the plate to them. If they are used it should be made clearer, if they aren’t reference to the IQIs should be removed.

The filtered images are examined using the CNR, COV and NNPS, which seems appropriate. However their effects are not very easy to see. I have examined both a printed version of figure 5 and the online version and I really can’t see the noise varying very much, not as much as the authors claim. Alternative images should be found where the noise is more obvious. Perhaps by zooming in more? I am sure there is something worth seeing as the results in figures 6 and 10 are far clearer.

Lines 187 to 195 appear out of place and should be removed (a second introduction?). A red arrow in figure 5 is mentioned on line 205, I think this should be a red line.

The results of the various noise filters are presented clearly in figure 8 and don’t need to be repeated in lines 220 to 229.

It is always difficult highlighting subtle variations in images and unfortunately they aren’t apparent in both figures 7 and 9 (as well as 5, as mentioned). The authors should consider alternative ways to present this data. Could the difference in the images compared to the unfiltered images be used? This may also reveal features around edges.

I hope this review doesn’t appear too critical as I think the work presented has merit and the subtle variations in this sort of image are always difficult to demonstrate. I would encourage the authors to consider alternative ways to present this. I also recommend a careful review of the English language as it is awkward and difficult to follow in places. And please be careful with the precision in your numerical results, which at times seems optimistic. I think some rounding is occasionally needed (e.g. line 285 “approximately 28.35 and 3.35”).

Round 2

Reviewer 1 Report

I think this revised version is well improved. A few comments are

  1. Figure 5 is still difficult to see any difference between TV and NLM in the pdf version of this manuscript. I suggest either have a much smaller ROI to zoom in or check if there’s any way to save the pdf in higher resolution. Because of now, this figure is basically no use and not supporting the conclusion. Maybe have a zoom-in in Figure 6 as well. It is not super clear that the NLM is better than TV in Figure 6 either.
  2. Line 98-99. Could you elaborate on how the RMSE is calculated with respect to Eq 11? Which one is I1 which one is I2, and what is the ROI? Would it be clearer if you have a table to list the results of the comparisons from simulations for RMSE and EPI?
  3. Why not change the contrast and brightness in figure 9 so that it gets easier to see the structures in the middle…
  4. I think the authors’ reply about the discussion about the difference between their NLM algorithm and the existing ones, can be nice to add to the introduction or discussion part.

Author Response

Dear reviewer,

Thank you for your useful comments and suggestions concerning our paper entitled “Effectiveness of non-local means algorithm with an industrial 3 MeV LINAC high-energy X-ray system for nondestructive testing”.

I think this revised version is well improved. A few comments are

  1. Figure 5 is still difficult to see any difference between TV and NLM in the pdf version of this manuscript. I suggest either have a much smaller ROI to zoom in or check if there’s any way to save the pdf in higher resolution. Because of now, this figure is basically no use and not supporting the conclusion. Maybe have a zoom-in in Figure 6 as well. It is not super clear that the NLM is better than TV in Figure 6 either.

√ Thank you for this comment. As you mentioned, the visually applied images of TV and NLM do not show a clear difference. Thus, it was corrected after supplementation using a smaller ROI as your recommendation to show the difference as much as possible.

  1. Line 98-99. Could you elaborate on how the RMSE is calculated with respect to Eq 11? Which one is I1 which one is I2, and what is the ROI? Would it be clearer if you have a table to list the results of the comparisons from simulations for RMSE and EPI?

√ Thank you for this comment.

  • The RMSE factor shows the relative difference between reference image (I1) and measured image (I2). Using the entire image as the ROI, the error between each pixel of I1 and I2 is calculated as the root-mean square. The smaller the RMSE value, the closer measured image is to the reference image. Also, the EPI factor did not show ROI because they were used as a reference for the whole area of image.
  • In addition, we added table for RMSE and EPI results as your recommendation.

  1. Why not change the contrast and brightness in figure 9 so that it gets easier to see the structures in the middle…

√ Thank you for this comment. We improved figure 9 as your recommendation.

  1. I think the authors’ reply about the discussion about the difference between their NLM algorithm and the existing ones, can be nice to add to the introduction or discussion part.

√ Thank you for this comment. We added sentences in the manuscript.

  • Added sentence: Especially, Xu et al [17] was introduced by modeling the nonlocal self-similarity (NSS) prior for noise reduction in natural image. However, its approach has limited to apply in the 2D X-ray image including the Poisson-gaussian mixed noise due to assume only considering the Gaussian noise. Since the Gaussian noise is independent of the imaging system, it is possible to artificially generate and learning various Gaussian noise in advance. On the other hands, the Poisson noise depends on the photon fluence (amount of photons). In addition, the X-ray image is expressed by superimposing from 3D to 2D, which has different characteristic from natural image. It is difficult to mathematically understand that there are many similar patches in an X-ray image notwithstanding which is overlapping and ambiguous borders compared to that of natural image.

The objective of this paper is to propose a NLM denoising algorithm and evaluate the image quality obtained with this algorithm using the industrial high-energy X-ray imaging system we established. Our proposed method is not required the additional manual work and a large amount of prior information during removing the mixed noise component, effectively.

Reviewer 2 Report

I would like to thank the authors for considering and acting on the recommendations that I made in my first review, and I would like to congratulate them on a significantly improved manuscript. The change in title is appropriate and the text as a whole flows much better and is easier to understand. I am particularly pleased with the updated results. Figures 4, 7 and 9 are helpful and show the value of the NLM algorithm.

Unfortunately the authors misunderstood my original comment that some rounding is occasionally needed (e.g. line 285 “approximately 28.35 and 3.35”). The point I was intending to make was the level of precision used in the paper was at times too high, and that some numbers should be rounded to fewer decimal places, not that approximations should be presented in quotation marks (“”). 28.35 does not seem approximate to me, it seems very precise. Therefore I suggest that the following minor changes be made (perhaps by the editorial office if allowed).

On the new manuscript:

Line 219 to now read: NLM algorithm was approximately 17.2, 4.5, and 3.4 times higher than that obtained with the

Line 222 to now read: approximately 28.4, 10.9, and 3.4 times smaller than that obtained with the noisy image, Wiener

Line 241/242 to now read: the proposed NLM algorithm is the lowest (approximately 10-10 mm2 at 2.0 lp/mm spatial frequency).

Line 261 to now read: and approximately 105 times lower than that of the noisy image. These results show that the

Line 284 to now read: was approximately 17.2 and 3.4 times higher than that of the noisy and TV images, respectively.

Line 285 to now read: Meanwhile, COV value of the NLM algorithm was approximately 28.4 and 3.4 times smaller

Line 287 to now read: 2 lp/mm was about 1 x 10-10 mm2, about 105 times lower than that for the noisy image. Our results

Once these very minor changes have been made I shall be happy that the manuscript will be ready for publication in Sensors.

Author Response

Dear reviewer,

Thank you for your useful comments and suggestions concerning our paper entitled “Effectiveness of non-local means algorithm with an industrial 3 MeV LINAC high-energy X-ray system for nondestructive testing”.

I would like to thank the authors for considering and acting on the recommendations that I made in my first review, and I would like to congratulate them on a significantly improved manuscript. The change in title is appropriate and the text as a whole flows much better and is easier to understand. I am particularly pleased with the updated results. Figures 4, 7 and 9 are helpful and show the value of the NLM algorithm.

Unfortunately the authors misunderstood my original comment that some rounding is occasionally needed (e.g. line 285 “approximately 28.35 and 3.35”). The point I was intending to make was the level of precision used in the paper was at times too high, and that some numbers should be rounded to fewer decimal places, not that approximations should be presented in quotation marks (“”). 28.35 does not seem approximate to me, it seems very precise. Therefore I suggest that the following minor changes be made (perhaps by the editorial office if allowed).

On the new manuscript:

Line 219 to now read: NLM algorithm was approximately 17.2, 4.5, and 3.4 times higher than that obtained with the

Line 222 to now read: approximately 28.4, 10.9, and 3.4 times smaller than that obtained with the noisy image, Wiener

Line 241/242 to now read: the proposed NLM algorithm is the lowest (approximately 10-10 mm2 at 2.0 lp/mm spatial frequency).

Line 261 to now read: and approximately 105 times lower than that of the noisy image. These results show that the

Line 284 to now read: was approximately 17.2 and 3.4 times higher than that of the noisy and TV images, respectively.

Line 285 to now read: Meanwhile, COV value of the NLM algorithm was approximately 28.4 and 3.4 times smaller

Line 287 to now read: 2 lp/mm was about 1 x 10-10 mm2, about 105 times lower than that for the noisy image. Our results

√ Thank you for this comment. I’m sorry confusion you. We modified sentences as your comments.

Once these very minor changes have been made I shall be happy that the manuscript will be ready for publication in Sensors.